

# Construction and Characterization of an Indoor Smog Chamber for the Measurement of the Optical and Physicochemical Properties of Aging Biomass Burning Aerosols Native to sub-Saharan Africa

5 **Damon M. Smith,** [1] **Marc N. Fiddler,** [2] **Kenneth G. Sexton**[3] **and Solomon Bililign**[2, 4]

[1]Energy and Environmental Systems Department, North Carolina A&T State University, Greensboro, North Carolina, USA
[2]NOAA-ISET Center, North Carolina A&T State University, Greensboro, North Carolina, USA
10 [3]Department of Environmental Sciences and Engineering, Gillings School of Global Public Health, University of North Carolina at Chapel Hill, NC, USA
[4]Department of Physics, North Carolina A&T State University, Greensboro, North Carolina, USA

*Correspondence to*: Solomon Bililign bililignsol@gmail.com

**Abstract.** We describe here the construction and characterization of a new smog chamber facility (NCAT chamber) for studying the chemical and optical properties of biomass burning (BB) aerosols from biomass fuels native to sub-Saharan Africa. This facility is comprised of a ~9 $m^3$ fluorinated ethylene propylene film (FEP) reactor placed in a temperature-controlled room and coupled with a cavity ring-down spectrometer, nephelometer, 20 condensation particle counter, differential mobility analyzer and other analytical instruments, such as $NO_X$ and $O_3$ analyzers, a GC, a filter sampler, and an impinger for collecting particles in water. Construction details and characterization experiments are described, including measurements of BB particulate size distribution and deposition rate, gas wall loss rates, dilution rate, light intensity, mixing speed, temperature and humidity variations, and air purification method. The wall loss rates for NO, $NO_2$, and $O_3$ were found to be $(7.40 \pm 0.01) \times 10^{-4}$, $(3.47 \pm 25\ 0.01) \times 10^{-4}$, and $(5.90 \pm 0.08) \times 10^{-4}$ $min^{-1}$ respectively. The $NO_2$ photolysis rate constant was $0.165 \pm 0.005$ $min^{-1}$, which corresponds to a flux of $(7.72 \pm 0.25) \times 10^{17}$ photons•nm•$cm^{-2}$•$s^{-1}$ from 296.0−516.8 nm. Particle deposition rate was found to be $(2.46 \pm 0.11) \times 10^{-3}$ $min^{-1}$ for pine at $D_p = 100$ nm. After initial mixing in the chamber, with the ultraviolet (UV) light off, the particle size distribution for BB samples used for the initial work did not stabilize until ~7.5 hours after injection peaking near a mobility diameter of ~340 nm. The chamber demonstrated gas and particle 30 loss rates, and other properties comparable to other similar indoor smog chambers

## 1. Introduction

Biomass burning (BB) is one of the largest sources of black carbon (BC), or absorbing aerosols, in the atmosphere, with as much as 30 % of aerosol mass belonging to BC (Andreae et al., 1998; Moosmüller et al., 2009; Bond et al., 2013). It is important to study BB aerosols because of their impact on air quality, cloud formation, 35 human health, and climate. While it is well known that BB aerosols contribute to climate forcing, much is still unknown about the extent of this forcing, owing to the high level of uncertainty regarding BB aerosol optical properties (Andreae and Merlet, 2001; Koch et al., 2009; IPCC, 2013). This uncertainty is partially due to the difficulty in quantifying the amount of light absorption and scattering by aerosols, which depends on their size,



chemical composition, age, humidity, and the wavelength of incident light (Harvey et al., 2016;Kim and Paulson, 2013;Kanakidou et al., 2005).

Biomass burning is a global phenomenon and the contribution of biomass burning aerosols in the atmosphere is the highest in the tropics (Shi et al., 2015). While the optical properties of BB aerosols produced by several North American biomass fuels have been and are being investigated, both in the laboratory and in field campaigns (Hodzic et al., 2007;Yokelson et al., 2009;Liu et al., 2014;McMeeking et al., 2009;Levin et al., 2010;Mack, 2008;Mack et al., 2010), BB aerosols for sub-Saharan African fuels were only investigated in field studies (Liousse et al., 2010;Formenti et al., 2003). To our knowledge, there have not been any laboratory studies of the optical properties of BB aerosols from fuels sources common in sub-Saharan Africa. Africa is the single largest continental source of BB emissions, with sub-Saharan Africa hosting some of the highest levels of atmospheric black and brown carbon (BrC) aerosols. Some studies estimate around 55 % of the global contributions to BB aerosols come from Africa (Ichoku et al., 2008;Roberts and Wooster, 2008;Roberts et al., 2009;Lamarque et al., 2010;van der Werf et al., 2010;Schultz et al., 2008).

There have been several studies on the physicochemical properties of BB particles in ambient investigations. Diameters and chemical compositions of wood-fire particles have been shown to change rapidly during the daytime, where photochemical oxidation increased the organic aerosol (OA) fraction (Zauscher et al., 2013). The cloud condensation nuclei (CCN) activity was found to be impaired by photochemistry (Engelhart et al., 2012;Giordano and Asa-Awuku, 2014). What happens to smoke particles in the absence of light and increased relative humidity (RH) is not yet clear (Murphy et al., 2006) and dynamic influence of humidity on the evolution of smoke particles is not defined (Li et al., 2015). Humidity may also change the physiochemical properties of aerosol during aging (Rubasinghege and Grassian, 2013). Understanding daytime and nighttime aging under the high RH conditions that are typical in the tropics is clearly needed. The influence of combustion properties, such as fire characteristics, burning stage, fuel type, and meteorological conditions on optical properties of BB aerosols for the region is also important.

The total mass of atmospheric aerosol is dominated by OA (20−90 %) and is composed primarily of secondary organic aerosol (SOA) (Zhang et al., 2011;Kanakidou et al., 2005;Jimenez et al., 2009;Gentner et al., 2017), which are produced from the atmospheric oxidation and processing of volatile organic compounds (VOCs) (Robinson et al., 2007;Gentner et al., 2012). A smaller fraction of OA is directly emitted as primary organic aerosol (POA) (Hallquist et al., 2009).

Quantifying the effects of net SOA formation and aging from BB smoke in the field poses a challenge because of variability between studies, due to different gas- and particle-phase emissions from different fuel types, burning conditions, external mixing of other aerosol types, and other environmental factors present in the region of smoke emission. A critical element of this uncertainty relates to the evolution of aerosol loadings within the atmosphere. In addition to an increase in overall aerosol loadings, secondary aerosol formation also acts to increase the single scattering albedo (SSA) of BB aerosol (Yokelson et al., 2009;Reid et al., 1998;Abel et al., 2003), potentially changing its climate forcing from a net warming effect to a net cooling effect. While SOA formation leading to a significant enhancement of OA mass has been widely observed in anthropogenic urban emissions, the



net contribution in aging biomass burning plumes remains unclear (De Gouw and Jimenez, 2009;Hawkins and Russell, 2010).

Laboratory studies are clearly needed to measure the optical properties of BB aerosols from African fuel sources as they age and interact with polluted air that has the same chemical profile as African mega cities. Smog
chambers provide a controlled environment to study the formation and the evolution of specific compounds and particles produced from specific fuel sources of interest by isolating the influence of emissions, meteorology, and mixing effects.

Smog chambers (both indoor and outdoor) have been used, beginning in the 1960s, to study atmospheric processes in controlled environments, such as biomass burning emissions (Kamens et al., 1984;Hennigan et al.,
2011;Bian et al., 2015), SOA production (Carter et al., 2005;Babar et al., 2016), diesel exhaust (Weitkamp et al., 2007), and cigarette smoke (Schick et al., 2012). As opposed to in situ measurements, the aging process can be studied over longer times in a chamber with greater temporal accuracy and sampling frequency.

Construction of smog chamber (inner reactor) walls can use a variety of materials, such as stainless steel (Akimoto et al., 1979;Wang et al., 2011) and Teflon FEP film (Cocker et al., 2001;Carter et al., 2005;Babar et al.,
2016). FEP film is the most widely used for large chamber designs due to its transparency in the ultraviolet (UV) spectrum, flexibility under changes in pressure, chemical inertness, and economic feasibility (Wang et al., 2014;Wu et al., 2007). While outdoor chambers provide the most realistic conditions when it comes to the solar spectrum, repeatability of experiments is poor, with too many changing variables dependent on the time of year, angle of the sun in the sky, and other weather conditions. Indoor chambers have the advantage of temperature and humidity
controls, as well as consistent light intensity (Babar et al., 2016). Modern ultraviolet bulbs, while not having the same spectra as solar light, can be characterized to create accurate models of atmospheric conditions. However, it is also worth noting that the difference in artificial light spectrum and solar spectrum may lead to different photolysis rates for some reactions (Carter et al., 2005;Takekawa et al., 2003).

In this paper, we describe construction and characterization of an indoor smog chamber built in our
laboratory at North Carolina A&T State University (hereafter referred to as NCAT Chamber) to study the optical and physicochemical properties of BB aerosols produced by fuel sources from sub-Saharan Africa. While studying these fuels is the goal of this facility, this paper will present work on white pine to facilitate a comparison with similar chambers. In addition to details of the construction and characterization experiments we describe how BB aerosols are produced and introduced into the chamber and how the aerosols are sampled for measuring their
broadband optical properties (absorption, extinction, and scattering) as a function of particle size and age.

## 2. Description of chamber and optical properties measurement facility

### 2.1. Chamber facility

The NCAT Chamber is located on one end of our 175 m$^3$ lab, on the opposite side of the room from the vents connected to the building's HVAC system. The chamber is placed as close as possible to test mixture
preparation sources (tube furnace and gas cylinders) and monitoring instruments to minimize injection and sample line losses, while allowing room for chamber operators to move between the chamber and surrounding instruments.



Mixture preparation sources are located inside or near laboratory hoods. Typical lab temperature is between 17 and 21 °C, with relative humidity between 30 % and 60 %. Under normal operating conditions, the chamber can remain near 0% humidity (below the detection limits of our instrument) with only minimal purge flow needed to obtain a

small positive pressure (balanced against air sampling requirements), having temperatures close to room temperature while the UV lights are off. With no flows to or from the chamber, humidity increases from near 0 % to under 7 % after 24 hours.

## 2.2. Chamber construction

The chamber system is composed of two parts: the inner reactor and the outer frame. The reactor is made of

12 rectangular frames connected by hex bolts to form a box having inner dimensions (length × width × height) of 2.36 m × 1.85 m × 2.06 m, for a volume ~9 m$^3$. Each frame is made of wood (having a cross section of 1.5 in by 1.5 in) and wrapped in fluorocarbon film (DuPont, type 500A FEP100, 50-inch width, 125 μm thickness), which is secured by staples to the back of the wooden frame. The frames are attached to each other so that the inner surface of the reactor is completely covered by the FEP film, leaving all wood surfaces facing the exterior, while the FEP

film itself is compressed between panels (using bolts approximately every six inches) to form an airtight seal. Some of the frames were reinforced by steel angle iron to straighten out the wood and provide a more rigid structure for the reactor. With this design, no heat-seals are necessary, which can be expensive and is prone to failure from flexing, which can result in leaking. With this design, the reactor was configured so that an individual panel could easily be detached from the chamber for either cleaning or repair. Cinder blocks are used to elevate the reactor to be

centered against the outer frame and to allow for airflow under the reactor for even temperature distribution. Two smaller steel panels were attached to the wooden frames on opposite sides of the reactor, flush against the FEP film to support stainless steel bulkhead fittings for injection and sampling lines. The FEP film, again, served as a seal for these fittings. While the structure of the reactor is rigid, the FEP film walls are flexible, always maintaining nearly constant room pressure in the chamber. Although it is assumed that the chamber is airtight, it is more likely that

there are small leaks in the chamber, which would allow room air to mix with chamber air. For this reason, the chamber is kept at a slight positive pressure during experiments to ensure that any leaks would be from the chamber into the room. In this way, chamber air does not become contaminated. Rising humidity in the chamber is one indication that room air is leaking into the chamber, due to the difference in relative humidity between chamber air and room air. If a leak becomes too large, or if the FEP film is punctured, the frame in question can be removed and

repaired.

The outer frame of the chamber is made of steel angle iron, having dimensions (length × width × height) of 2.55 m × 2.45 m × 2.48 m, which is primarily used for mounting the UV lights and protective coverings. On each of two opposite sides of the chamber, there are two rows of eight light fixtures spaced evenly along the side of the chamber. Each fixture holds two bulbs (Sylvania, 30-watt UVA, 356 nm centered spectrum, F30T8/350BL/ECO,

36"), for a total of thirty-two bulbs to a side, or sixty-four bulbs for the chamber. Each side of the chamber (hereafter referred to as door side or window side, corresponding to the door and window of the laboratory) has been wired to a separate circuit with a light switch, allowing for irradiation using either 50 % or 100 % of the total light intensity


available. Due to the amount of space available in our lab, UV lamps are ~18 cm from the FEP film surface. Chamber temperature is elevated slightly during operation of the UV lamps and is described in more detail below.

Protective Masonite is attached to the outer frame on all sides of the chamber, including ceiling, except for two windows surrounding the injection and sampling panels which are left uncovered for visual inspection of the chamber and the floor where the Masonite is attached directly to the wooden frame of the reactor. The inner surface of the Masonite is covered with aluminum foil to reflect and scatter most of the light emitted from the UV lamps surrounding the chamber to simulate uniform irradiance.

**2.3. Air purification**

Room air can be drawn into the chamber through a 3-inch diameter PVC pipe, fitted with a dust filter to keep out contaminants and dust. The laboratory exhaust system is connected to the chamber via a check-valve and gated valve used as the exhaust port. This serves to evacuate the chamber with either the room air for primary flushing or with clean air for thorough chamber cleaning. This room air input is primarily used to quickly evacuate

the chamber after a burn has been completed. To generate clean air, compressed house air, having very low humidity, is passed through a 0.01 μm activated carbon filter (Speedaire, 4GNN4), followed by a zero-air generator (Aadco Instruments, 747-30), and a ballast tank (Aadco Instruments). The generator is capable of a maximum flow rate of 30 L/min and removes any particulates, hydrocarbons, $NO_x$, and $SO_x$ from the air, while the ballast tank's volume keeps pressure fluctuations, and, by extension, flow rate fluctuations, to a minimum. Gas purity is less than

1 ppb for $O_3$ and CO and less than 0.5 ppb for $SO_2$, $H_2S$, NO, $NO_2$, $NH_3$, and both methane and non-methane hydrocarbons. Air flow to the chamber is controlled by a high-flow rotameter (Cole-Parmer EW-32033-16), which is capable of flow rates greater than 60 L/min and is used to balance flows going into the chamber with flows coming out of the chamber. Since the rigidity of the chamber frame keeps the FEP film walls from completely collapsing in on itself, flow rates going into and out of the chamber must remain balanced (or slightly positive, as

stated above). While zero-air is sent directly to the chamber after the ballast, air used for other systems is pressure regulated with a single stage regulator (Grainger, 4ZM13).

**2.4. Gas and particle injection**

Additional gases, such as commercially prepared gas cylinders of NO and $NO_2$, in nitrogen can be injected via a second, low-flow rotameter (Cole-Parmer EW-32033-16) with a maximum flow of 1.25 L/min. Concentrations

can be calculated from timed injections, certified reagent gas concentration, flow-rate, and chamber volume. A three-way valve between the rotameter and chamber input can be set to exhaust, chamber, or off. In the exhaust position, the flow rate can reach equilibrium before switching to the chamber position, allowing for accurate measurement of injections if elapsed time is known. Other chemical compounds, such as aromatic hydrocarbons (high purity), can be injected in the liquid phase by syringe through a rubber stopper into a glass U-tube. If

necessary, these chemicals can be evaporated by heating the U-tube. A portion of the air flow from the zero-air generator can be diverted through the U-tube to flush the chemicals into the chamber. Gases injected in this manner are used to simulate atmospheric conditions. A mixing fan is mounted below these inputs on the inside the reactor,



which generally produces a well-mixed volume within 15 to 20 minutes after an injection.

For laboratory samples, BB aerosols are generated by combusting wood samples in a tube furnace. The furnace (Carbolite Gero, HST120300-120SN) holds an 85 mm OD, 80 mm ID, and 750 mm long quartz working tube and has a heated region of 300 mm. Stainless steel mounts and insulation plugs on either end enable the introduction and sampling of gasses. Air flow through the furnace is regulated by two mass flow controllers (MFC, Sierra Instruments), one each for clean air and nitrogen, with a combined flow rate of 10 L/min. In this way, the oxygen concentration available for combustion can be altered to study both oxygen-rich and oxygen-depleted

environments. Clean air is supplied by house compressed air that passed through the zero-air generator, while nitrogen comes from a compressed cylinder (Airgas/National Welders, UN1066, industrial grade nitrogen). The furnace is preheated to 500 °C before the introduction of biomass samples. These samples are placed in a quartz combustion boat (AdValue Technology, FQ-BT-03), weighed and pushed into the center of the furnace with the aid of tongs, before replacing the input insulator and flange. The smoke and gasses produced from combustion are sent

directly to the chamber via heated (200 °C), and ¼ inch OD stainless steel transfer tube. After introduction, they undergo cooling and dilution in a similar fashion to natural processes; as opposed to step-wise dilution and cooling, which introduces a hysteresis in the condensation of semi-volatile species.

**2.5. Particle sampling and aerosol optical measurements**

BB particles are taken from the FEP environmental chamber via graphite impregnated silicone tubing

before entering a 710 μm impactor inlet (3.8 μm diameter cut point), neutralizer, and long differential mobility analyzer (DMA) (TSI, Model 3080), where the aerosol is size selected. Flow through the entire system (0.58 sL/min) is produced by a pump within the condensation particle counter (CPC), and the DMA sheath flow is 2.8−6.0 L/min in single blower mode. Aerosol flow then enters a ring-down cavity, where the aerosol extinction is measured over a range of wavelengths. The extinction coefficient $\alpha_{ext}$ ($m^{-1}$) is measured by the cavity ring-down system

(CRDS), and the extinction cross section $\sigma_{ext}$ ($m^2$/particle) is found using $\alpha_{ext}$ and the number density of particles $N_{CRD}$ (particles/$cm^3$) in the cavity. After the CRDS, aerosol scattering coefficients are measured at 453, 554, and 698 nm using the integrating nephelometer (TSI, model 3563), with particle concentration measured by the water-based condensation particle counter (WCPC, TSI, model 3788). Collectively, we have dubbed this system the Aerosol Optical Properties Measurement (AOPM) setup, and it has been extensively characterized (Singh et al., 2014). The

absorption cross section is derived from the difference between the scattering and extinction cross sections. SSA is found by dividing the scattering by the extinction coefficient and adjusting for the relative number density in each instrument. Alternatively, the CPC can be attached directly after the DMA, where it is used to collect size distribution spectra of the biomass burning aerosol (i.e. as a scanning mobility particle sizer (SMPS)). SMPS spectra are corrected for multiple charging effects using built-in software algorithms. Additional sampling lines are used to

measure ozone, nitric oxides, carbon monoxide, carbon dioxide, and hydrocarbons using the analyzers listed in Table 1. Measurements of NO, $NO_2$, and $O_3$ are calibrated before and after a set of experiments, and those calibration factors are averaged to determine resulting concentrations. Figure 1 provides schematics of the chamber and the full experimental set up. Sampling lines are also used to collect particulates via impinger (AGI-4 type, Ace



Glass Inc., 7540-04 and 7542-10) and filters (Tisch Environmental, SF18040, 2-µm pore). The impinger was initially developed to capture BB aerosol in situ, keeping the sample viable until it can be re-aerosolized in the laboratory (Singh et al., 2016). However, the size distribution of the impinged aerosol is different from fresh BB aerosol directly sampled from the chamber, as shown in Fig. 2. Additionally, recent work in our group has shown that retrieved refractive index values and the wavelength dependence of cross sections are altered by impinging and re-aerosolizing (Poudel et al., 2017). It is thought that most of the organic compounds are dissolved in the water and that particles were left leaving little to no coating when the samples are re-aerosolized. It has been shown that most of the BB aerosol material (on average about 50 %) is highly refractory organic material that is soluble in water (Mayol-Bracero et al., 2002), which leaves a nearly spherical compact core that produced a shift towards smaller mobility diameters in the size distribution. However, the size distribution of the impinged samples remains stable for an extended period, as shown in Fig. 2. Filter samples have been imaged by a scanning electron microscope (SEM) to determine their morphology for modeling refractive indices and optical properties using the T-matrix code (Poudel et al., 2017), and will be used in future studies to determine morphology of aged African BB particles.

## 3. Chamber characterization experiments, results, and discussion

The main limitation of chamber experiments is that particles and gas-phase species may be lost to chamber walls on timescales that are shorter than the timescales of the atmospheric processes being studied (Bian et al., 2015). Therefore, before use, chambers must be characterized to reduce uncertainties caused by the loss of gasses and particles in the chamber walls, as well as "background reactivity" which includes possibly surface chemistry which can generate gases, including reactive radical species (Carter et al., 2005;Weitkamp et al., 2007). Both particle-phase and vapor-phase wall losses must be considered in chamber experiments involving BB.

Typical chamber losses include dilution, vapor and particle deposition, gravitational settling, collisions with the walls of the chamber, agglomeration of smaller particles into larger ones, and leaks from the chamber (Crump and Seinfeld, 1981;Kamens et al., 1984). Particle loss rates are generally lower for larger chambers due to a smaller surface-area-to-volume ratio. Contamination was a large source of error in early experiments, leading to evacuable chamber designs. It is now common practice to condition chamber walls, especially FEP walls, by flushing the chamber with pure air and irradiating it to remove volatile particles (Akimoto et al., 1979;Grieshop et al., 2009). Off-gassing of $NO_x$ and other contaminants can also increase uncertainty for low sample concentrations (Carter et al., 2005;Wang et al., 2014). Proper chamber evacuation or flushing is key to removing contaminants left behind after each experiment. Ozone, hydrogen peroxide, and nitrogen oxide are also sometimes used to react with these residual species in order to maintain chamber cleanliness (Babar et al., 2016).

The characterization experiments conducted for the NCAT chamber are described below and results are compared with similar chambers.

## 3.1. Chamber mixing, flushing, and cleaning

A fan (12VDC spark free marine in-line electric blower (Shurflo, Yellow 3")) is used to thoroughly mix chamber injections with chamber air. The fan is rated to 3.40 $m^3$/min and was used at 25−33 % of maximum power





by adjusting a variable voltage regulator. The injections of NO and $NO_2$, were made using ~54 and 80 ppm,
certified, of $NO/NO_2$ (Airgas/National Welders) respectively at 1 L/min for 20 min. Figure 3 shows four injections
with 5 minute durations (for a total of 20 minutes of injection), spaced out every 20−30 minutes. Overall injection
volume/concentration is the same for all experiments. They become well mixed after 15 minutes, as shown in Fig. 3.
This includes the time it takes for particles to stabilize. The actual mixing time is nearly the same as the mixing time
reported by other indoor chambers (Babar et al., 2016;Wang et al., 2014), which exhibit mixing times of 2−3
minutes. Based on the available literature, the mixing time seems to be independent of the size of the chamber. The
mixing fan rotation speed has been shown to provide better mixing but causes an increase in wall loss (Carter et al.,
2005;Wang et al., 2011). Particles, however, do not become well mixed (i.e. have a relatively stable size
distribution) until 60–90 minutes. This may be due in part to rapid agglomeration, coagulation, and particle growth.

After an experiment, the chamber is flushed with air from the zero-air generator at 30 L/min. An equal flow
leaves the chamber through a gate valve and exhaust line that connects to the duct of the fume hood. While the vent
line is equipped with a fan that is identical to the mixing fan, it is not typically used. At this rate, it usually takes less
than 48 hours to remove all trace gases to reach acceptable zero values. Irradiation is not currently being used to
clean the chamber since no contaminants are found after 48 hours. Particulates can take up to 72 hours to remove
completely at the same flow rate. Particle removal was monitored using the DMA and the chamber was considered
clean when particle concentrations were lower than the limit of detection; hence the longer time for particle removal
from the chamber.

### 3.2. Wall loss rates of gases

Adsorption of gases on the chamber walls is the primary cause for the decrease of gas concentrations in the
chamber. Wall loss rates of NO, $NO_2$, and $O_3$ were evaluated by injecting $0.12 \pm 0.01$ ppm of NO and $0.18 \pm 0.01$ ppm
of $NO_2$ during separate experiments and continuously monitoring their decay in the dark and near zero RH, as shown
in Fig. 4. While the wall loss rate of $O_3$ was similarly calculated, the initial concentration of ozone injection is
unknown due to the large volume of our chamber and the small amount of $O_3$ produced by the lamp (Spectroline,
11SC-1) inside our ozone generator. Ozone injections took over three days to generate high enough concentrations
in the chamber to measure the wall loss rate. It is likely that the injection rate was only slightly higher than the
dilution rate. Wall loss rates are obtained by considering the wall loss as a first order process. To determine the total
decay of a species in the gas phase, the natural log of its relative concentration is plotted over time, as shown in Eq.
(1)

$$Ln\left(\frac{x(t)}{x(0)}\right) = -k_{tot}t \qquad\qquad 1$$

where $x(0)$ is the initial concentration of a species, $x(t)$ is its concentration at a given time $t$, and $k_{tot}$ is the total decay
rate constant. When plotted this way, $k_{tot}$ is the negative of the slope. The dilution rate constant, $k_d$, for each species
can be determined by summing the sampling flow rates $f$ for the chamber and dividing them by the chamber volume
$V$:



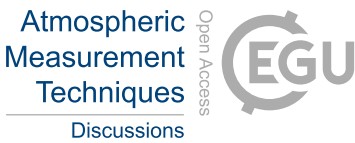

$$k_d = \frac{\sum_i (f_i)}{V}$$  2

Since this factor is independent of concentration, it can be used for all species. It is then subtracted from $k_{tot}$ to get

the wall loss rate, $k_{wl}$, for the species:

$$k_{wl} = k_{tot} - k_d$$  3

This factor includes losses due to gravitational settling, impaction with the walls, and leaks from the chamber. For the NCAT chamber loss rate constants for NO, $NO_2$, and $O_3$ are $(7.40 \pm 0.01) \times 10^{-4}$, $(3.47 \pm 0.01) \times 10^{-4}$, and $(5.90 \pm 0.08) \times 10^{-4}$ $min^{-1}$, respectively. Table 2 shows comparisons with similar chambers which represents averages of

several measurements. The errors are one standard deviation ($1s$) of these repeat measurements. Minimizing surface area to volume ratios, and attempting to isolate the sampled flow from the walls help reduce losses, but challenges remain (Bruns et al., 2015). However, our results are slightly faster than chambers with lower surface area to volume ratios. A possible reason for this is that gas transfer towards the walls are driven by the turbulence inside the chamber (La et al., 2016), which is a result of fan position and speed. Since, fan speeds were not provided for all

chambers, wall loss comparisons become difficult.

### 3.3. Particle wall loss rate and shift in size distribution

Biomass burning particles produced in a tube furnace were introduced into the chamber as described in section 2.4. To enable the comparison with other chambers, white pine wood was chosen. It was weighed to ~100 mg and placed in a quartz combustion boat which was in turn placed at the center of the furnace. After the initial

combustion of biomass, multiple peaks are seen in the size distribution of the resulting BB aerosol. Both the number of peaks and the diameters they correspond to shift apparently at random as the particles are being mixed inside the chamber. After ~60−90 minutes, the size distribution resolves into a single log-normal distribution with a peak particle mobility diameter near 246 nm for pine. The total particle concentration was ~$1.5 \times 10^5$ particles•$cm^{-3}$ after mixing. Even under dark conditions (i.e. the UV lamps are off and near zero RH), this distribution continues to shift

towards larger particle sizes, reaching peak mobility diameters of ~340 nm after 7.5 hours. This changing distribution can be seen in Fig. 5.

As with the gas wall loss rates, particle wall loss, or deposition, follows the same first-order process, and is dependent on the concentration and particle size. It is given by Eq. (4) (Cocker et al., 2001):

$$Ln\left(\frac{N(D_p,t)}{N(D_p,0)}\right) = -k_{dep}(D_p)t$$  4

where $N(D_p,t)$ is the number concentration in particles•$cm^{-3}$ with diameter $D_p$, and $k_{dep}(D_p)$ is the wall loss rate constant of particle with diameter $D_p$. While the rate constant varied over time, it was consistent during the beginning, middle and end of the experiment. For particle diameters of 100 nm, a rate constant of $(2.46 \pm 0.11)$ x





$10^{-3}$ min$^{-1}$ was found once the size distribution stabilized after ~7.5 hours. The particle wall loss rates decreased

with increasing particle diameter suggesting smaller particles deposit faster (Wang et al., 2014;Takekawa et al., 2003). For $D_p$ = 100 nm the particle wall loss rate for NCAT chamber is comparable to other chambers reported for example 3.83 x $10^{-3}$ min$^{-1}$ for the Guangzhou (30 m$^3$) chamber (Wang et al., 2014) and 3.96 x $10^{-3}$ min$^{-1}$ for the 7 m$^3$ KNU chamber (Babar et al., 2016). A smaller value of 7.5 x $10^{-4}$ min$^{-1}$ was reported for the 29 m$^3$ Ilmari chamber (Leskinen et al., 2015). This is much lower than most reported particle wall losses. There were no mixing

fans in the Ilmari chamber, which was used to explain the difference. Mixing fans can enhance wall deposition by increasing turbulence inside the chamber. The wall loss rate was faster initially and leveled off after 10 hours in the NCAT chamber. Similar observations were reported in the Ilmari chamber (Leskinen et al., 2015).

It is likely that the size distribution changes due to a competition between wall loss and particle coagulation, with subsequent formation of larger particles. When modeled as a second order process, a poor

relationship between $1/N$ (100 nm,$t$) and $t$ was found. However, for polydisperse aerosol number loss (i.e. when the total particle volume concentration (nm$^3$•cm$^{-3}$) was plotted), a good fit was found for a first order decay. The loss rate constant for this process was (1.34 ±0.02) × $10^{-3}$ min$^{-1}$ which corresponds to a life time of 12.4 hours. If one assumes that particle volume is conserved (i.e. that condensation or evaporation of semivolatile species is either very fast or absent), a better fit would be expected. These results are comparable to other chambers, which were 2.8 x

$10^{-3}$ min$^{-1}$ for the Guangzhou chamber (Wang et al., 2014) 3.48 x $10^{-3}$ min$^{-1}$ for Paul Scherrer Institute, (Paulsen et al., 2005) and 1.5 x $10^{-3}$ min$^{-1}$ for the California Institute of Technology Chamber (Cocker et al., 2001). More experiments are planned to understand the relationship between initial particle number concentration and the loss rate.

Recent experiments have demonstrated that POA from biomass burning contains semi-volatile compounds,

which can evaporate during dilution (May et al., 2013). Particle wall losses have been used to quantitatively correct observed aerosol concentrations to deduce SOA formation in smog chambers (Hennigan et al., 2011) by assuming that semi-volatile compounds are in equilibrium with the particles deposited in the walls instead of the walls themselves. The magnitude of vapor wall loss has not been considered in most smog chamber studies, since the values are not constrained (Bian et al., 2015). Recently, Zhang et al. (2014) showed that the vapor wall loss can

underestimate SOA formation. It is therefore necessary to consider vapor wall loss when introducing BB smoke in chambers. We did not consider vapor wall loss in this work since the focus was on POA.

Particle and vapor losses to the walls of a smog chamber during an experiment can alter the composition of the smoke being studied, causing great uncertainty for quantifying SOA production (Weitkamp et al., 2007). In one study with cigarette smoke, particle deposition to chamber walls ranged from 23 % to 78 %, which is dependent on

the wall material and independent of particle concentration and flow rate (Schick et al., 2012). Other studies simulated the chamber wall loss of particles and vapors from BB smoke (Bian et al., 2015;Trump et al., 2016). Bain et al. found that 41 % of the initial particle-phase organic mass was lost during the experiments, with 65 % of this loss from direct particle deposition and the remaining 35 % from evaporation of OA and subsequent loss of vapor to the walls. The later mechanism is influenced by dilution during the introduction of particles into the chamber. The

shift in the size distribution to larger mobility diameters can also be explained by smaller particles diffusing towards





the chamber walls more quickly and coagulating more rapidly, as was observed by Wang et al. (2014) and Takekawa et al. (2003).

Particle loss rates were also studied in detail by Pierce et al. (2008). Experiments involving wood combustion generally produce BB aerosols in the range of 70−230 nm in mobility diameter, with most of the
360 particles having a mobility diameter of less than 500 nm (Kamens et al., 1984). This shift can be lessened by using total particle concentrations of less than 25,000 particles•cm$^{-3}$ (McMurry and Grosjean, 1985). We plan to study size distribution of photochemically aged BB aerosol in greater detail as part of our ongoing research goals.

### 3.4. Light intensity and temperature

The UV lamps (Sylvania, F30T8/350BL/ECO, 36") used for the NCAT chamber simulate the UV portion
of the solar spectrum, as shown in Fig. 6. For this range, the FEP film is over 90 % transparent (DuPont, 2010). Average light intensity for the chamber is modeled using the photolysis rate of $NO_2$, which is estimated by a steady-state actinometry (Wang et al., 2014). The photolysis rate constant $j_{NO2}$ is calculated from the measured concentrations of NO, $NO_2$, and $O_3$ after the lights are turned on and the reaction reached equilibrium, and is given by Eq. (5)

$$j_{NO_2} = \frac{k_{NO+O_2}[O_3][NO]}{[NO_2]}$$

where [NO], [$O_3$], and [$NO_2$] represent concentrations (molecule•cm$^{-3}$), and $k_{NO+O3}$ is the rate constant of $NO_2$ formation from ozone and NO, which is $k_{NO+O3} = 1.9 \times 10^{-14}$ cm$^3$•mol$^{-1}$•sec$^{-1}$ (Burkholder, 2015). The average photolysis rate constant for our chamber is $0.165 \pm 0.005$ min$^{-1}$ at full light intensity, as shown in Fig. 7. The
375 maximum $NO_2$ photolysis rate in our chamber is comparable to those reported in other indoor chambers, such as 0.132 min$^{-1}$ in the 29 m$^3$ Ilmari chamber (Leskinen et al., 2015); 0.12 min$^{-1}$ in the 27 m$^3$ chamber at the Paul Scherrer Institute (Paulsen et al., 2005); 0.19 min$^{-1}$ in the 90 m$^3$ environmental chamber at the University of California, Riverside (Carter et al., 2005); 1.5 min$^{-1}$ in the 28 m$^3$ chamber at the California Institute of Technology (Cocker et al., 2001); and comparable to most recent chambers provided in Table 2. When only half of the lights
were powered, values of $0.091 \pm 0.005$ min$^{-1}$ and $0.0846 \pm 0.0042$ min$^{-1}$ were obtained for the door side and window side, respectively. The lower rate constant for the window side may be due to the observation windows built into the chamber, which are located near the window side and may allow light to escape.

The effect of excluding the rate of $NO_2$ removal on $j_{NO2}$ was also examined. Since the rate constant for the total loss of $NO_2$ is relatively small ($4.06 \times 10^{-4}$ min$^{-1}$), $j_{NO2}$ only decreased by 0.24–0.30 % when this factor was
385 included. As such, while this is not significant for the NCAT chamber, it should be considered for chambers with lower light intensity or when a chamber is sampled at flow rates that are high relative to the chamber volume.

The actinic flux of the chamber was also measured (Hamilton et al., 2011). The spectrum of the UV lamps, the measured $j$ value for $NO_2$ under several conditions, and the quantum yield and absorption cross section for $NO_2$ (Burkholder, 2015)were used to determine the actinic flux in the region where these lights would be





photochemically active towards $NO_2$, which is 296−424 nm. In this method, the spectrum was integrated using the trapezoidal method and normalized to 1 photons•cm$^{-2}$•s$^{-1}$ over the 296−424 nm range. When both banks are engaged, a flux of $(7.42 \pm 0.24) \times 10^{17}$ photons•nm•cm$^{-2}$•s$^{-1}$ was found over this wavelength range. When taken over the entire range of the spectrometer (296.0−516.8 nm, the flux becomes $(7.96\pm0.25) \times 10^{17}$ photons•nm•cm$^{-2}$•s$^{-1}$. The flux for the bank of lights near the door was $(4.22\pm0.24) \times 10^{17}$ and $(4.52\pm0.26) \times 10^{17}$

photons•nm•cm$^{-2}$•s$^{-1}$ over 296−424 nm and 296.0−516.8 nm, respectively. Similarly, the set near the windows were $(3.92\pm0.18) \times 10^{17}$ and $(4.20\pm0.20) \times 10^{17}$ photons•nm•cm$^{-2}$•s$^{-1}$ for their respective ranges.

Indoor smog chambers present a potential fire hazard due to the heat generated by the UV lamps and trapped inside the reflective walls of the chamber. With the lamps on, the maximum temperature recorded in the NCAT chamber was under 30 °C after 5 hours of use, as shown in Fig. 8. The typical change in temperature is less

than 10 °C after 5 hours, with most of the increase occurring in the first hour of operation.

### 3.5. Soot generation and characterization

Most laboratory studies using smog chambers utilize large combustion chambers (Bian et al., 2015) or residential wood stoves (Grieshop et al., 2009) to generate BB aerosols from combustion. Others use hoods to sample smoke in a more ambient environment (Singh et al., 2016). While these methods are useful in determining

the broader impact of BB, they possess too many poorly constrained variables. Even when a single fuel type is used, there remain contaminants in the surrounding air. Additionally, different parts of the same fire may be undergoing different burning stages, which have distinctly different combustion profiles. To avoid these issues, we have employed a tube furnace, described in greater detail in section 2.4, which allows us to combust a measured amount of biomass at a specific temperature to more accurately investigate variations in burning stages. In addition to

temperature control, this design also allows us to vary the oxygen available for combustion, to investigate smoke produced from oxygen-starved burns. This control over the combustion process will allow us to accurately measure the contribution of specific species (sources) to the aerosol mass loading of the environment.

Most indirect (and some direct) aerosol optical measurement techniques involve collecting the samples on filters before they can be analyzed. However, the use of filters can affect these measurements due to multiple

scattering by the filter itself, the spatial distribution of the particles on the filter, and changes in the morphology of these particles upon deposition (Bond and Bergstrom, 2006;Moosmüller et al., 2009). Further, samples collected on filters are more prone to artifacts due to their highly-concentrated nature (Moosmüller et al., 2009). Direct measurements can be taken using a combination of techniques that maintain the state of the aerosol, such as one of the following methods: photoacoustic spectroscopy (PAS) and integrating nephelometry to measure absorption and

scattering, respectively, with extinction being the sum of the two; or CRDS and integrating nephelometry to measure extinction and scattering, respectively, while absorption is determined from extinction minus scattering. We use the extinction-minus-scattering method in our lab. For highly absorbing particles, such as BC, this method is quite reasonable (Bond and Bergstrom, 2006), as it has been shown elsewhere (Lewis et al., 2009;Langridge et al., 2013) that PAS can include large errors under certain conditions. Additionally, our CRDS uses tunable lasers that provide

a wide range of solar wavelengths (Singh et al., 2016), while most studies are limited to a single or a few specific





wavelengths. The setup in our laboratory will allow us to sample particles directly from the chamber, and measure their size distributions and optical properties as a function of aerosol age.

## 4. Conclusion

We have shown here that the new NCAT indoor smog chamber can simulate atmospheric conditions for studying radiative impacts of BB aerosols. The chamber demonstrated gas and particle loss rates, and other properties comparable to similar indoor smog chambers. Novel aspects of this chamber include the use of a tube furnace and the combustion of small quantities of fuel under controlled conditions of temperature and gas (nitrogen and air) flow, which enables the simulation of different burning stages and measuring of the combined combustion efficiency of each burning stage in a controlled manner. The chamber is connected to a SMPS to determine the size

distributions of fresh and aged BB aerosols as they form and change in size over time. BB particles are sampled directly into a cavity ring-down spectrometer and a nephelometer for measuring their scattering and extinction cross sections using tunable lasers that provides a wider range of optical wavelengths than previously reported. This characterized and calibrated system will determine the effects of fuel type, aging, and humidity on the size distribution, optical properties, and physicochemical properties of biomass burning aerosols native to sub-Saharan

Africa for the first time.

## 5. Acknowledgements

The authors acknowledge Marquin Spann for his experimental assistance and the financial support provided by the National Science Foundation under grant # NSF-AGS 1555479 to accomplish this work.




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





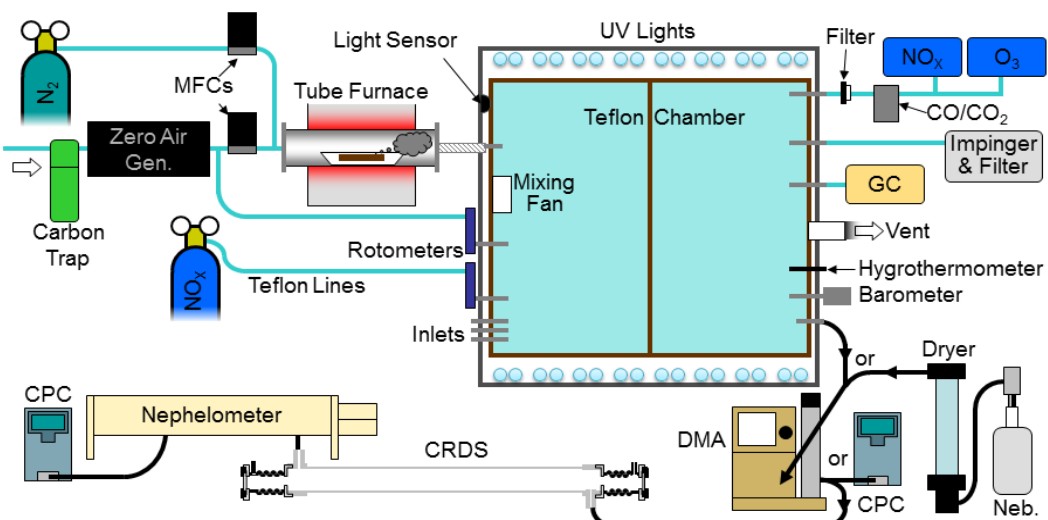

**Figure 1: Schematics of the NCAT Chamber with all inputs and sampling lines, including analytical instruments and optical properties measurement facility (lasers not shown).**

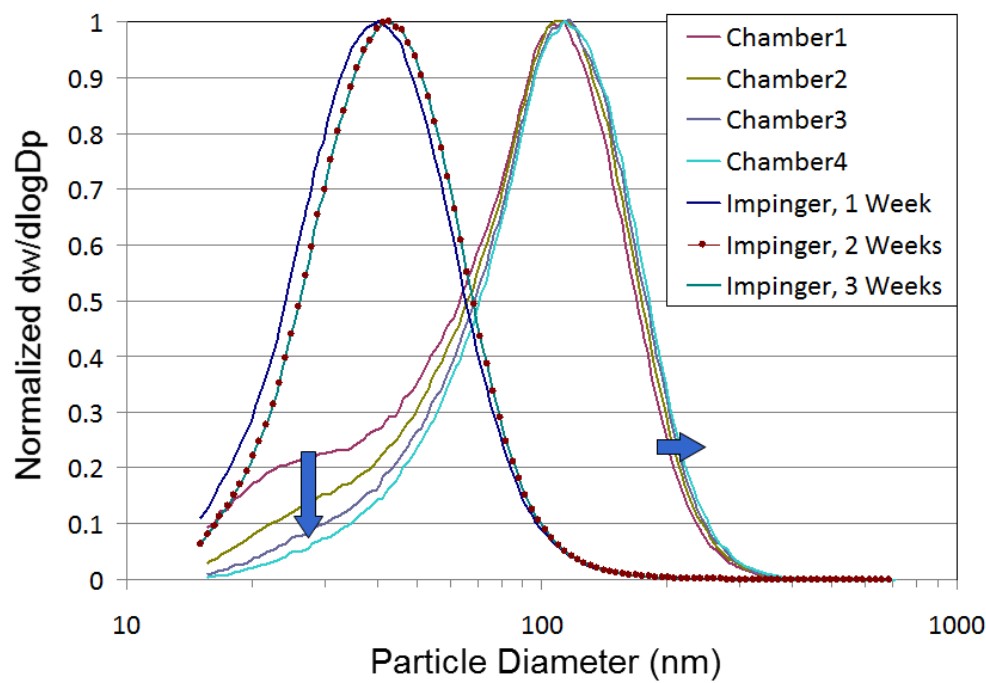

**Figure 2: Normalized size distributions of fresh BB aerosol, measured directly from the chamber (right), compared to re-aerosolized BB aerosol, collected via impinger and stored in water (left). Note the shift and stability of the size distribution of impinged samples.**



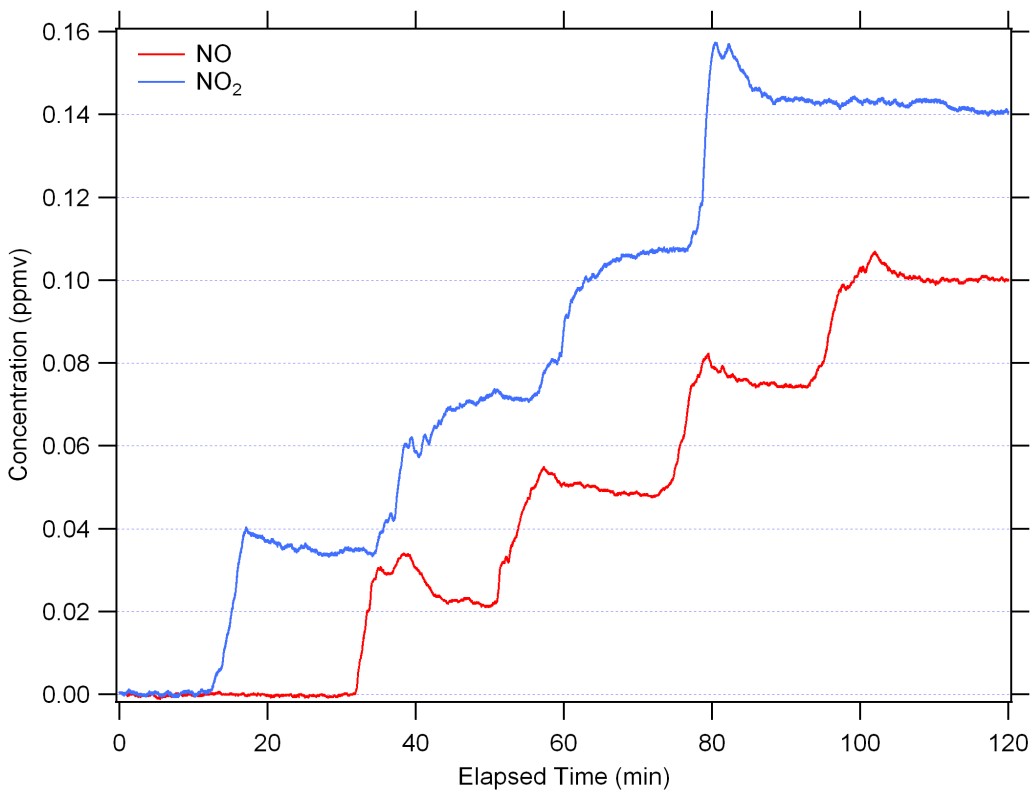

**Figure 3: Injections and response (mixing) time for NO and NO₂, done on different days.**

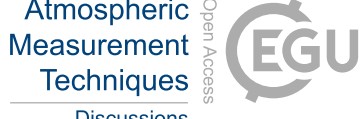

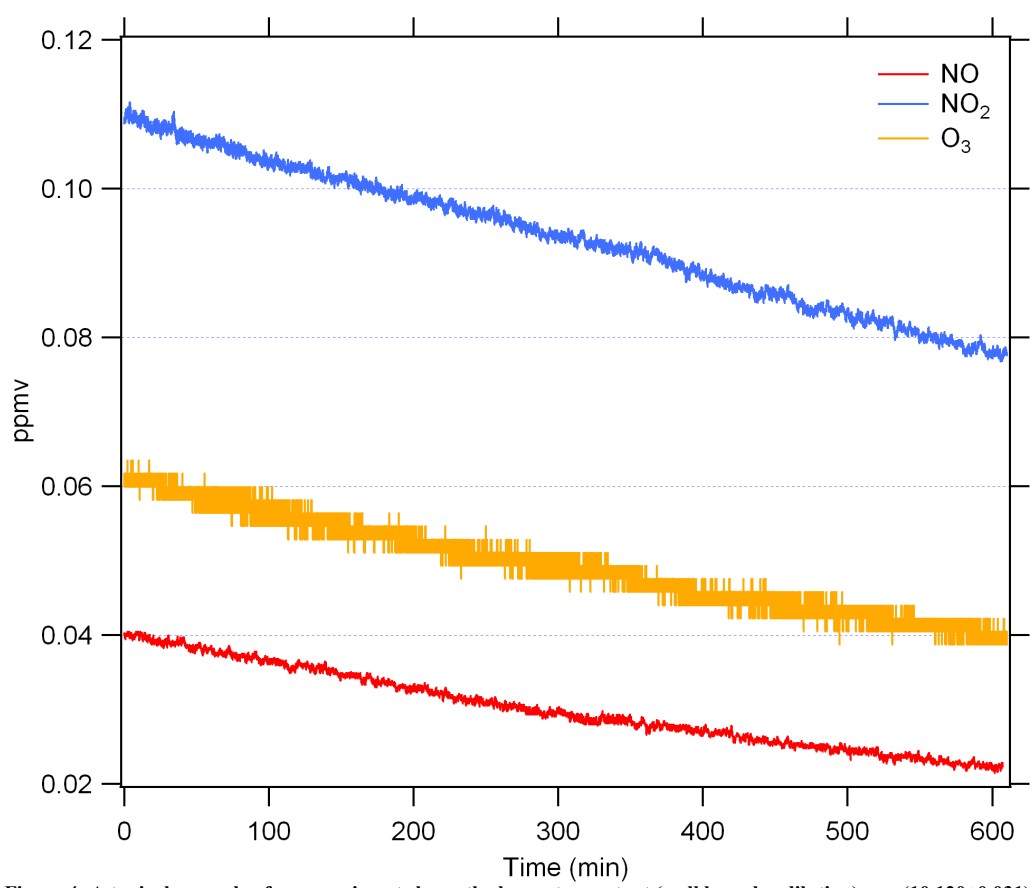

**Figure 4: A typical example of an experiment shows the loss rate constant (wall loss plus dilution) was $(10.120\pm0.031) \times 10^{-4}$, $(7.796\pm0.033) \times 10^{-4}$, and $(6.487\pm0.164) \times 10^{-4}$ $min^{-1}$ for NO, $NO_2$, and $O_3$, respectively. This corresponds to lifetimes of 16.67, 21.31, and 25.69 hours, respectively. The results in Table 2 are averages of several runs.**




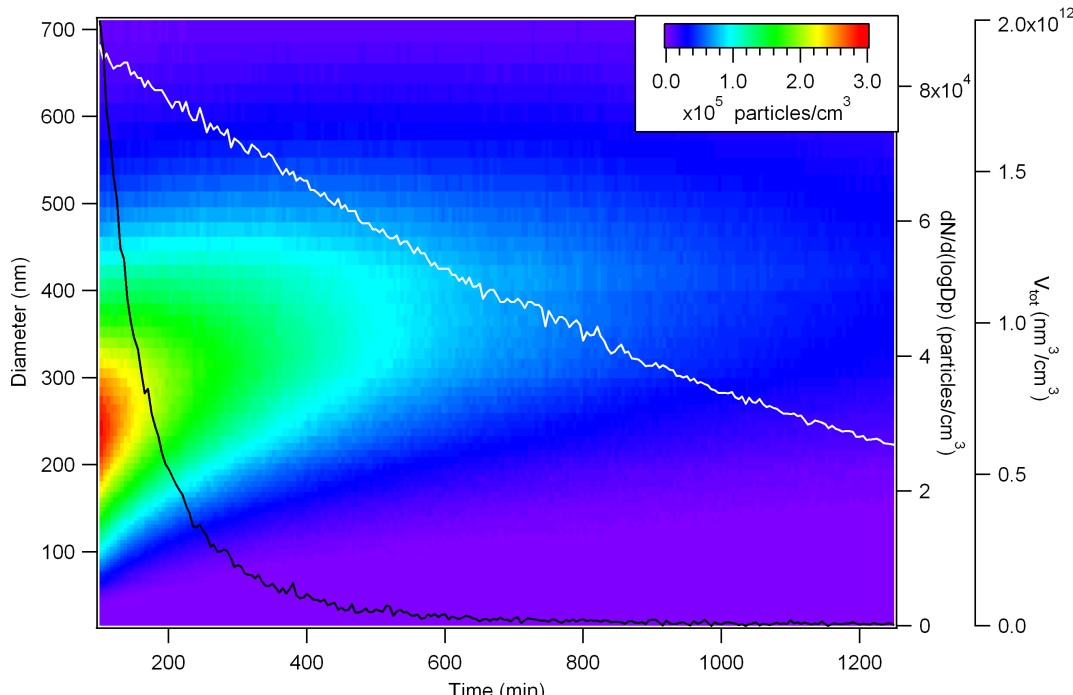

**Figure 5: The number density size distribution of pine combustion over time (background) with particle loss at $D_p$ = 100 nm (black) and aerosol volume loss (white). The line at t -450 shows where the size distribution stabilizes**





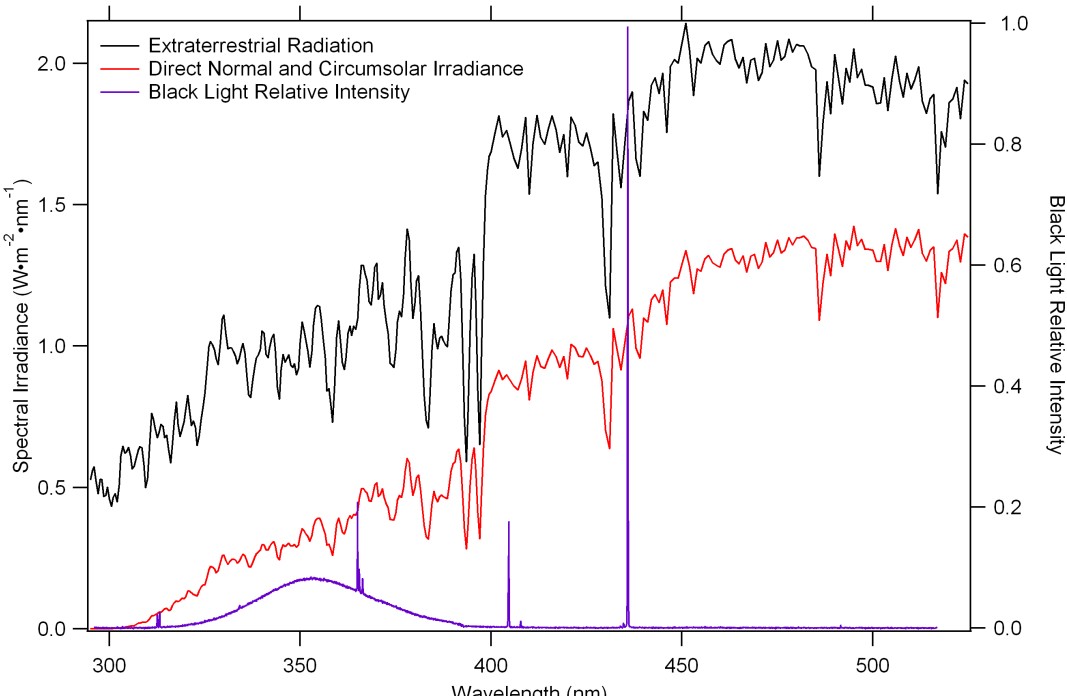

**Figure 6: Comparison of the solar spectrum to the UV lamps in our chamber. Extraterrestrial radiation (black) and direct normal and circumsolar irradiance (red) is from the ASTM G173-03 reference spectra derived from SMARTS v. 2.9.2 (RReDC). UV lamps (blue) were measured in our laboratory using Ocean Optics spectrometer (HR2000+ GRATING #H12, with wavelength range 296.0−516.8 nm, a ILX-511B detector with UV window, and 10 microns slit width.**



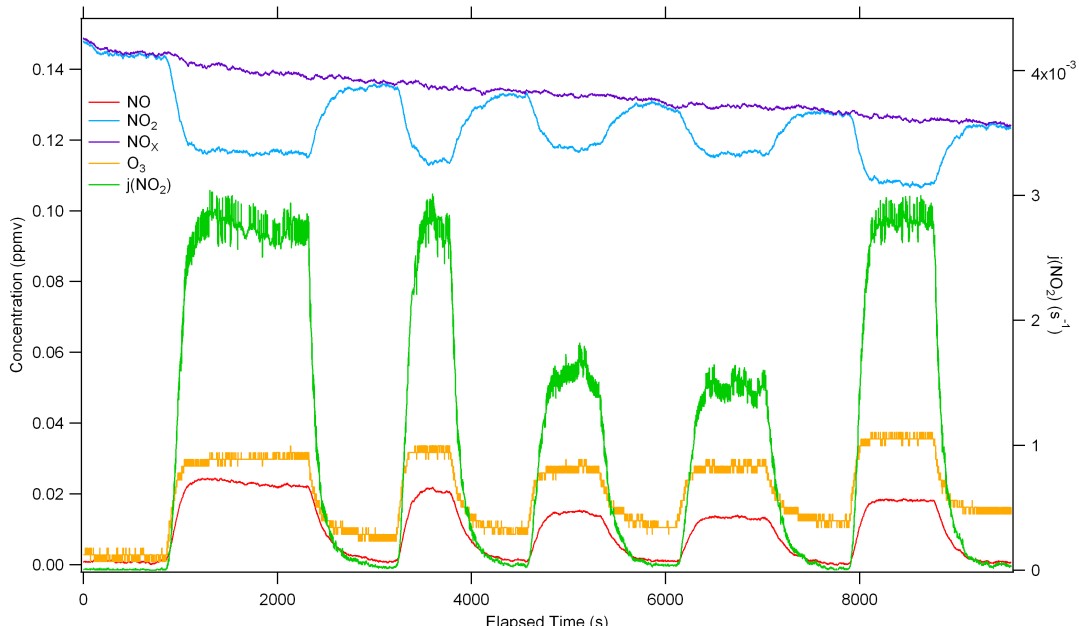

**Figure 7: A plot of the NO$_2$ photolysis rate constant ($j$) against concentrations of NO, NO$_2$, and O$_3$. The largest first, second, and last $j$ peaks correspond to all lights turned on (100 %), with $j$ values of (2.73±0.09) × 10$^{-3}$ s$^{-1}$, (2.78±0.11) × 10$^{-3}$ s$^{-1}$, and (2.78±0.07) × 10$^{-3}$ s$^{-1}$, respectively. The third and fourth peaks represent the door side and window side of the chamber, respectively, and only use half of the lights (50 %), with $j$ values of (1.56±0.09) × 10$^{-3}$ s$^{-1}$ and (1.45±0.07) × 10$^{-3}$ s$^{-1}$, respectively.**



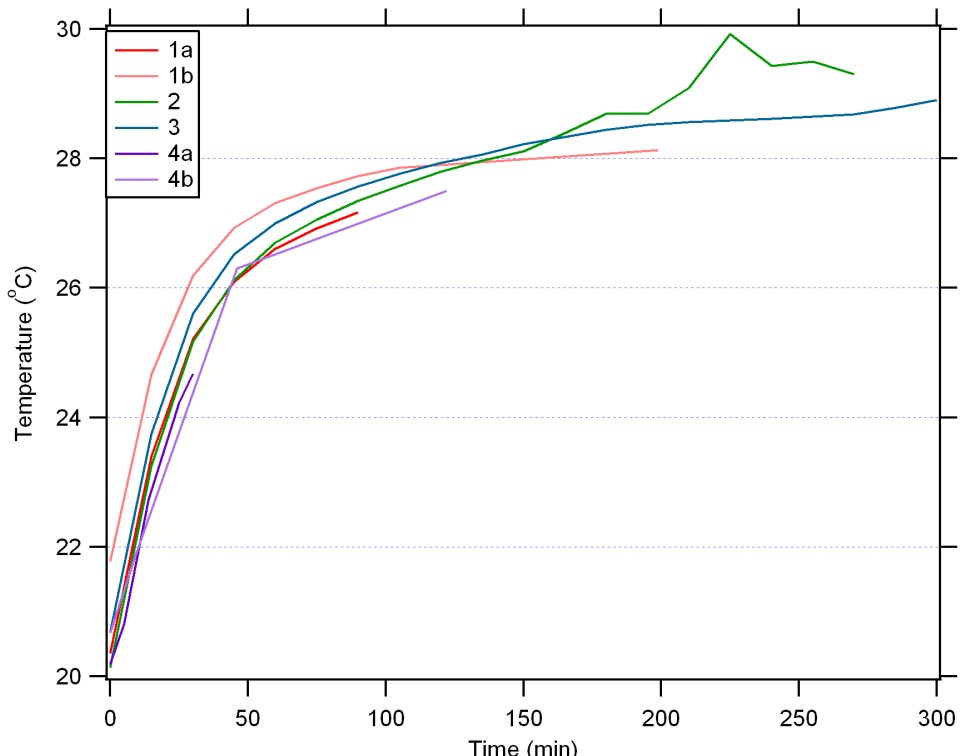

**Figure 8: Temperature profile of the chamber. Runs 1b and 4b were done on the same day as runs 1a and 4a, respectively, with initial temperatures higher for 1b and 4b than 1a and 4a.**



**Table I. List and description of analytical instrumentation used with the NCAT chamber.**

| Species/Variable | Instrument | Range | Detection Limit | Accuracy |
|---|---|---|---|---|
| NO/NO$_2$ | Nitrogen Oxides Analyzer, Monitor Labs, Model 8340, San Diego, CA, USA | 0–1 ppm | 2 ppb | ±1 ppb |
| O$_3$ | U.V. Photometric O$_3$ Analyzer, Thermo Electron, Model 49, Franklin, MA, USA | 0–1 ppm | 2 ppb | ±2.5 ppb |
| Hydrocarbons | Analytical Gas Chromatograph, Carle Instruments, Model 211 | 0–1 ppm typical | 2–10 ppb | ±2 to 10 ppb |
| CO$_2$/CO | Portable IAQ Meter, AZ Instrument, Model 77597, Taichung City, Taiwan | CO$_2$: 0–5000 ppm CO: 0–1000 ppm | 1 ppm | CO$_2$: ±30 ppm ±5 % of reading CO: ±10 ppm (≤ 100 ppm), ±10 % (between 101–500 ppm), ±20 % (> 500 ppm) |
| Pressure | Omega Instruments, Part #PX309-015AI, Norwalk, CT, USA | -15 to 50 psig | N/A | ±0.25 % BSL, max |
| Temperature/Relative Humidity/Dew Point | Traceable Products, Model 4085, Webster, TX, USA | Temp: -40 °C to 104.4 °C RH: 5–95 % | Temp: 0.01 RH: 0.01 % | Temp: ±0.4 °C (0–40 °C), ±1 °C otherwise RH: ±1.5 % |



**Table 2. Chamber characteristics and decay rates in min⁻¹ for several comparable indoor atmospheric chambers. KNU: Kyungpook National University (Babar, 2016). TU: Tsinghua University (Shan, 2007). GIG-CAS: Guabgzhon Institute of Geochemistry-Chinese Academy of Science (Wang, 2014). Ilmari University of Eastern Finland (Leskinen, 2015).**

| Chamber | Volume $m^3$ | Surface to Volume Ratio $m^{-1}$ | NO $min^{-1}$ | $NO_2$ $min^{-1}$ | $O_3$ $min^{-1}$ | $k_{dep}$ (100 nm) $min^{-1}$ | $j_{NO2}$ $min^{-1}$ |
|---|---|---|---|---|---|---|---|
| NCAT | 9.0 | 2.90 | $(7.40\pm0.01) \times 10^{-4}$ | $(3.47\pm0.01) \times 10^{-4}$ | $(5.91\pm0.08) \times 10^{-4}$ | $(2.46\pm0.11) \times 10^{-3}$ | $0.165\pm0.005$ |
| KNU | 7 | 3.09 | $3.78 \times 10^{-4}$ | $4.48 \times 10^{-4}$ | $6.47 \times 10^{-4}$ | $3.96 \times 10^{-3}$ | 0.17 |
| TU | 2 | 5 | $3.38 \times 10^{-4}$ | $4.17 \times 10^{-4}$ | $6.07 \times 10^{-4}$ | $2.6 \times 10^{-3}$ | 0.23 |
| GIG-CAS | 30 | 2.07 | $1.41 \times 10^{-4}$ | $1.39 \times 10^{-4}$ | $1.31 \times 10^{-4}$ | $3.83 \times 10^{-3}$ | 0.49 |
| Ilmari | 29 | 1.97 | | | | $7.5 \times 10^{-4}$ | 0.132 |