# Peer review of "Construction and Characterization of an Indoor Smog Chamber for the Measurement of the Optical and Physicochemical Properties of Aging Biomass Burning Aerosols Native to sub-Saharan Africa"

_Atmospheric Measurement Techniques, 2018_

## Referee Comment (RC1) · Anonymous Referee #3 · 31 Mar 2018

Smith et al. describe the construction and characterization of an indoor chamber for aging studies of biomass burning aerosols native to sub-Saharan Africa. Given the importance of biomass burning aerosols in regional and global environmental issues, the subject matter is somewhat appropriate for AMT. However, the paper cannot be accepted in its current form given its lack of novelty, insufficient results and ambiguous writing. The paper should only be accepted if these issues are addressed.

General comments:

1. I fail to see what makes this indoor chamber novel, and how this chamber is different from the numerous indoor smog chambers already constructed and are currently being used. The authors need to be explicit when describing the novelty of their chamber.

2. The authors claim that that this chamber is constructed specifically for aging studies of biomass burning aerosols native to sub-Saharan Africa. However, it seems like this chamber can be used to study any biomass burning aerosol system, not necessarily those native to sub-Saharan Africa. Does the combustion of biomass burning aerosols native to sub-Saharan Africa require special experimental apparatus or setup that is different from those currently being used in other chamber facilities? This is not clear from the manuscript. This also makes the title of the manuscript very deceiving. The title "Construction and characterization of an indoor chamber for the measurement of the optical and physicochemical properties of aging biomass burning aerosols" seems more appropriate.

3. The amount of results related to the characterization of the indoor chamber presented in this manuscript is insufficient, especially those related to particle wall loss. This makes it hard to judge if the chamber is truly suitable for biomass burning aerosol aging studies as claimed by the authors.

4. I find the flow of the manuscript meandering and somewhat confusing. There are parts of the manuscript (specifically the results section) where I fail to see how the paragraph(s) is related to the point the authors are trying to make.

Specific major comments:

1. Line 112: How does the lab's temp and RH affect the chamber's temp and RH (when the chamber lights are off)? It would be useful to have a figure showing this. This figure will also help

convince readers that the lab's large RH range (30 to 60 %) will not affect the experimental conditions in the chamber.

2. Line 232: Is the chamber operated as a "batch reactor" or as a "continuous flow reactor"? This should be stated in the manuscript.

3. Line 254: What are the concentrations of NO and NO2 added into the chamber based on the flow rates and concentrations of certified NO/NO2 gas cylinders? Providing these numbers will make it easier to compare these concentrations to those measured by NO or NOx monitors shown in Fig. 3. On a related note, does "elapsed time" in the x-axis of Fig. 3 refer to the time passed since the first gas injection? This should be stated explicitly in the text. Also, please indicate in Fig. 3. when the additional gas injections were made.

4. Line 262: The authors state that particles only become well-mixed in their chamber after 60 to 90 min. A figure showing how the aerosol number/volume concentration changes as a function of elapsed time after aerosol injection into the chamber should be used to back up their claim.

5. Line 278: The authors state that O3 injections took over 3 days. This sentence does not make sense. Assuming that the chamber was run as a "batch reactor" in this experiment, I would expect their 9 m3 chamber to be completely deflated by day 3, even with minimal chamber leaks. If bag deflation did not take place, then what is the volume of the chamber on day 3? On a related note, the initial O3 concentration injected into the chamber can be estimated from the O3 injection time into the chamber and the O3 production rate of the lamp.

6. Line 303: The authors claim that biomass burning aerosols from white pine wood was used to enable comparison with other chambers. It is difficult to make direct comparisons when all the chambers used in this comparison use different aerosols to characterize their chambers. For example, the Cocker and Wang papers use ammonium sulfate seed aerosols to characterize particle wall loss rates in the Caltech and Guangzhou chambers, respectively. On a related note, it will be easier for the authors to characterize particle wall loss rates in their chamber if they used inert aerosols such as ammonium sulfate (similar to the Cocker and Wang papers) instead of biomass burning aerosols. This is because inert aerosols will not be subjected to changes in particle size as a result of gas-particle partitioning processes (unlike the biomass burning aerosols used by the

authors), thus making the calculation of particle wall loss rates for the different particle size bins easier.

7. Line 308: What are the initial aerosol volume and surface area concentrations used in these experiments?

8. Line 317: The sentence "While the rate constant varied over time, it was consistent during the beginning, middle and end of the experiment" does not make sense. I would expect a single rate constant for each particle size bin.

9. Line 319: The observation that "The particle wall loss rates decreased with increasing particle diameter…" should be demonstrated using a figure showing how the particle wall loss rates change with particle size (similar to the one shown in Fig. 1 of Loza et al. (2012)).

10. How does the particle wall loss rates change with chamber RH and temperature? It will be worth knowing this, especially in the case of temperature since the authors showed in Fig. 8 that the chamber temperature can increase by ~9 C during a 5 h experiment when the chamber lights are switched on.

11. Line 321: It seems more appropriate to compare particle wall loss rates at different particle sizes for the different chambers, as opposed to at just one particle size (100 nm).

12. Were routine particle wall loss experiments performed (e.g., weekly, monthly experiments)? How reproducible are the particle wall loss rates for the different particle sizes?

13. Lines 329 to 334: The discussion about modeling particle wall as a second order process should be supported with figures showing the fits.

14. Line 336: Previous papers have already determined that coagulation plays an important role when large aerosol concentrations are used. See Nah et al. (2017).

15. There should be some discussion on how the authors plan to correct for particle wall loss in their experiments. For example, do they plan to use size-dependent particle wall loss rates (see Loza et al. (2012), Nah et al. (2017)), or the average loss rate of the total aerosol mass or number concentration (see Carter et al. (2005), Pathak et al. (2007), Pierce et al. (2008))?

16. Lines 339 to 356: These two paragraphs don't add any value to the discussion. They seem to belong in the intro.

17. Line 358: Pierce et al. (2008) is not the only paper that studied particle wall loss rates (see references in point 13). Please cite accordingly.

16. Line 401: No results were presented in section 3.5 (Soot generation and characterization). I don't see the point of this section.

Minor comments:

1. Line 86: Should be "As opposed to field measurements"

2. Line 102: Should be "white pine wood"

3. Line 389: There should be a space after (Burkholder, 2015). Also, it should be "Burkholder et al."

4. Fig. 2: The legend is ambiguous. What do "Chamber1, Chamber2, Chamber3 and Chamber4" in the legend refer to?

References:

Carter, W. P. L., Cocker, D. R., Fitz, D. R., Malkina, I. L., Bumiller, K., Sauer, C. G., Pisano, J. T., Bufalino, C., and Song, C.: A new environmental chamber for evaluation of gas-phase chemical mechanisms and secondary aerosol formation, Atmos. Environ., 39, 7768–7788, doi:10.1016/j.atmosenv.2005.08.040, 2005.

Loza, C. L., Chhabra, P. S., Yee, L. D., Craven, J. S., Flagan, R. C., and Seinfeld, J. H.: Chemical aging of m-xylene secondary organic aerosol: laboratory chamber study, Atmos. Chem. Phys., 12, 151–167, doi:10.5194/acp-12-151-2012, 2012

Nah, T., McVay, R. C., Pierce, J. R., Seinfeld, J. H., and Ng, N. L.: Constraining uncertainties in particle-wall deposition correction during SOA formation in chamber experiments, Atmos. Chem. Phys., 17, 2297–2310, https://doi.org/10.5194/acp17-2297-2017, 2017.

Pathak, R. K., Stanier, C. O., Donahue, N. M., and Pandis, S. N.: Ozonolysis of alpha-pinene at

atmospherically rele- vant concentrations: Temperature dependence of aerosol mass fractions (yields), J. Geophys. Res.-Atmos., 112, D03201, doi:10.1029/2006jd007436, 2007.

Pierce, J. R., Engelhart, G. J., Hildebrandt, L., Weitkamp, E. A., Pathak, R. K., Donahue, N. M., Robinson, A. L., Adams, P. J., and Pandis, S. N.: Constraining particle evolution from wall losses, coagulation, and condensation-evaporation in smog- chamber experiments: Optimal estimation based on size distribution measurements, Aerosol Sci. Tech., 42, 1001–1015, doi:10.1080/02786820802389251, 2008.

---

## Referee Comment (RC2) · Anonymous Referee #1 · 12 Apr 2018

This study describes a smog chamber facility that is designed for studying the physicochemical and optical properties of biomass burning aerosols. Characterization experiments, including measurements of particle size distribution, gas and particle wall loss rates, chamber dilution rates, light intensity, and chamber mixing timescale, are presented. These characterization experiments have been routine practices in the majority of laboratory chambers around the world that are employed to study atmospheric chemistry. The authors did not bring any new scientific advances but simply repeated a selection of well-established experimental procedures. I would not recommend publication on AMT in the current form. Significant revisions are needed, as described in the individual points below.

General:

The authors constantly claim the novelty of their newly constructed chamber, yet fail to provide any experimental evidence that could help to identify any unique aspect of the current chamber setup. What new discoveries could potentially be produced by the NCAT chamber compared with all the other chambers that are also equipped to study BB aerosols? One novel aspect of this chamber, as the authors state in the conclusion section, is 'the use of a tube furnace …., that enables the simulation of different burning stages …'. However, the authors did not provide any experimental observations in terms of the physical, chemical, and optical properties of BB aerosols produced from different burning stages of a given fuel. Relevant measurements need to be added in the manuscript to support this claimed novelty of the chamber.

While the title highlights that the NCAT chamber is particularly suitable to study the aging processes of BB aerosols, the authors emphasize their focus on primary organic aerosols throughout of the main text. One illustration experiment that shows how photochemical aging would affect the optical properties of BB aerosols needs to be provided.

The authors spend an entire paragraph in the introduction section discussing the need to investigate the effect of relative humidity on the evolution of BB aerosols. Can relative

humidity be well controlled in the NCAT chamber? If so, the humidity effect on the SMPS measured size distribution of BB aerosols needs to be given.

A large body of discussions in the main text, such as air purification, chamber flushing and cleaning, light spectra shown in Figure 6, and temperature profiles shown in Figure 8, have been well established routine chamber operation procedures for many years and can be moved to Supplement. Experimental evidence needs to be given in Section 3.5 to support the statement 'the setup in our laboratory will allow us to sample particles directly from and chamber, and measure their size distributions and optical properties as a function of aerosol age'. As suggested earlier, one experiment that illustrates changes in the optical properties of BB aerosols as a function of photochemical aging needs to be given.

Specific:

Page 3, Line 87: Most static Teflon chambers are actually not ideal for studying aerosol aging. As the experiment proceeds, the Teflon bag volume will be continuously depleted, eventually leading to significant particle wall losses. A typical chamber experiment usually lasts for 24-36 hours, which equals to approximately one to two days of atmospheric OH exposure given the average OH concentration of ~$10^6$ molecules cm$^{-3}$ generated in the chamber.

Page 5, Line 165: Measurements need to be provided in the Supplement to support the statement 'gas purity is less than …and both methane and non-methane hydrocarbons'.

Page 7, Line 228-230: Have the authors tried to vary the atomizing pressure or use other solvents (e.g., acetonitrile) to dissolve impinger collected BB aerosols to see if these procedures would make a difference on the measured aerosol size distribution?

Page 8, Line 275-280: How was the onset of gas chamber losses defined here? It seems like over the entire course of gas chamber loss measurements, standard gases are continuously injected into the chamber and withdrawn out of the chamber with a balanced flow rate. If so, uncertainties caused by chamber mixing timescale need to be considered. To fully isolate the effect of dilution on the measured gas wall loss rate, the authors are

suggested to inject standard gases, turn off the injection flow and wait till well mixed, and then start the wall loss measurement.

Page 10, Line 338: Initial particle number concentration might be an indicator of the particle pysicochemical properties, e.g., phase state. Understanding the relationship between initial particle number concentration and the loss rate needs to be included in the current study.

Page 10, Line 346: When BB aerosols produced from a 500 °C furnace are introduced into a chamber operated at room temperature, organic vapors generated together with BB aerosols will undergo condensation on existing particles. On the other hand, organic compounds may evaporate from BB aerosols upon dilution in the chamber. So strictly speaking, the focus here is not POA.

Page 23, Figure 5: Please also provide the measured particle wall loss rate as a function of the particle diameter.

Page 27, Table 1: What type of detector is attached to GC for the identification of hydrocarbons?

---

## Author Comment (AC1) · 13 May 2018

**RESPONSES TO REVIEWER #3**

The authors thank the reviewer for his detailed and comprehensive comments that will help make the paper stronger. We will address all the comments and suggested corrections as best as we can, below. Note: Figure numbers are consecutive for the new figures provided for the responses. For a figure from the submitted manuscript the same number is used.

**General comments:**

**REVIEWER COMMENT #1**: I fail to see what makes this indoor chamber novel, and how this chamber is different from the numerous indoor smog chambers already constructed and are currently being used. The authors need to be explicit when describing the novelty of their chamber.

AUTHOR RESPONSE: The reviewer is right in indicating that there are numerous indoor smog chambers already constructed; however, we do not claim that the chamber itself is novel. What we consider unique and novel is the "Combustion-Chamber System". Given the unique combination of our custom designed and built instruments and combustion aerosol generation system, our entire chamber system is better described as "Unique". The following are unique qualities of our system: BB combustion and aerosol generation using a tube furnace to produce aerosols under controlled burning conditions (temperature, air flow, oxygen content, and amount of fuel burned) and the ability to clearly visually differentiate brown carbon (often formed at around 450-500 °C), Black Carbon (formed around 650 - 750 °C) and other forms (mixed brown/black carbon) at intermediate temperatures. Figure 1 shows two forms of carbon collected on filters. The generation and introduction of the soot particles into the chamber is described in line 194-197. The generation of soot particles and the introduction method, along with filter and impinger sampling and the integration of the chamber to the cavity ringdown/nephelometer system are not common arrangements, and we haven't come across chambers used in this manner. Many chambers are difficult to clean and therefore the characterization for wall loss and light intensity is a moving-target. Our chamber can more easily have each of the FEP panels cleaned and replaced. We believe the entire system is novel, not just the chamber. There is more to say here about the "novel" nature of the chamber system: We have characterized this chamber with particles that are combustion particles from biomass burning and have expended efforts to fully document the characterization with this manuscript. This adds to the "novel" or more rare nature of the chamber in that many chamber characterization efforts are undocumented, which makes the chamber results suspect and uncertain. Other chambers have been characterized with simple experiments, which are incomplete or unsatisfactory for conducting experiments later with different particles, such as from biomass burning. These aspects of our chamber and Figure 1 will be included in the revised manuscript.

Figure 1. Filter samples of Biomass Burning aerosols produced in a tube furnace (a) at 450 °C b) at 800 °C. (Note the 800°C sample is highly concentrated)

**REVIEWER COMMENT #2**: The authors claim that that this chamber is constructed specifically for aging studies of biomass burning aerosols native to sub-Saharan Africa. However, it seems like this chamber can be used to study any biomass burning aerosol system, not necessarily those native to sub-Saharan Africa. Does the combustion of biomass burning aerosols native to sub-Saharan Africa require special experimental apparatus or setup that is different from those currently being used in other chamber facilities? This is not clear from the manuscript. This also makes the title of the manuscript very deceiving. The title "Construction and characterization of an indoor chamber for the measurement of the optical and physicochemical properties of aging biomass burning aerosols" seems more appropriate.

**AUTHOR RESPONSE**: We agree with the reviewer that the title gives the impression that a special chamber is needed to study African fuels. Indeed, we don't need a special chamber for African fuels. However, the reason for including that in the title is to emphasize the fact that, to our knowledge, there have not been any laboratory chamber studies of biomass aerosols produced from African biomass fuels. We already have preliminary measurements of optical properties of biomass burning aerosols from fuels native to the region (see below for responses to comment #22). However, we have no objection in changing the title to "Construction and Characterization of an Indoor Smog Chamber for the Measurement of the Optical and Physicochemical Properties of Aging Biomass Burning Aerosol."

**REVIEWER COMMENT #3**: The amount of results related to the characterization of the indoor chamber presented in this manuscript is insufficient, especially those related to particle wall loss. This makes it hard to judge if the chamber is truly suitable for biomass burning aerosol aging studies as claimed by the authors.

**AUTHOR RESPONSE**: We have demonstrated that 100 nm particles (those with the shortest chamber lifetime, and far smaller than we plan to use for optical characterization) persist for sufficient periods to enable their collection and/or optical measurement. We have not presented size-dependent wall loss rates constants in this work, nor have many others. The stated goal of

this work is optical property characterization. If we were to determine SOA yield or combustion emission factors, such size-dependent wall loss rate constants would be necessary, and we would perform that level of characterization should we expand to those goals. However, it is not necessary at this time for our purposes. In short, there is no step in our data analysis where a particle wall loss rate constant would be used, so measurement of size-dependant wall loss rate constants are not currently useful. However, the comments may be significant if only the reviewer explained why he thinks the amount of characterization results are insufficient. The reviewer has not specifically stated what he feels is missing from the manuscript.

**REVIEWR COMMENT #4**: I find the flow of the manuscript meandering and somewhat confusing. There are parts of the manuscript (specifically the results section) where I fail to see how the paragraph(s) is related to the point the authors are trying to make.

**AUTHOR RESPONSE**: Though it is not clear what the reviewer means by this comment, we will conduct a careful review of the paper to maintain a logical flow of ideas and communicate clearly.

**Specific major comments**

**REVIEWER COMMENT #5:** Line 112: How does the lab's temp and RH affect the chamber's temp and RH (when the chamber lights are off)? It would be useful to have a figure showing this. This figure will also help convince readers that the lab's large RH range (30 to 60 %) will not affect the experimental conditions in the chamber.

**AUTHOR RESPONSE**: As stated in lines 134 - 139: Although the chamber is air tight in theory, in fact there are most likely small leaks. These leaks would allow ambient (room) air into the chamber when positive pressure is not maintained. As a result, chamber humidity increases as the purity of chamber air decreases. Lines 113-117 state that the temperature in the chamber is close to room temperature with UV lights off. While the low rate of sampling and replenishment is maintained, chamber RH is independent of its surroundings, since the air provided by the house compressor that is further purified by the zero-air generator and our conditioning system is extremely dry ( $\sim$ 0%).Room temperature varies by only a degree or two during chamber operation.

**REVIEWER COMMENT #6:** *Line 232: Is the chamber operated as a "batch reactor" or as a "continuous flow reactor"? This should be stated in the manuscript.*

AUTHOR RESPONSE: The chamber is operated as a "batch reactor" having a fixed volume of 9 m3. The Teflon walls are somewhat flexible, allowing for small changes in volume during injections without a change in pressure. This will be stated in the revised manuscript. It is not a "continuous flow reactor", which the authors take to mean a device in which photochemical age is related to the position within the reactor.

**REVIEWER COMMENT #7:** *Line 254: What are the concentrations of NO and NO2 added into the chamber based on the flow rates and concentrations of certified NO/NO2 gas cylinders?*

Providing these numbers will make it easier to compare these concentrations to those measured by NO or NOx monitors shown in Fig. 3. On a related note, does "elapsed time" in the x-axis of Fig. 3 refer to the time passed since the first gas injection? This should be stated explicitly in the text. Also, please indicate in Fig. 3. when the additional gas injections were made.

**AUTHOR RESPONSE**:**

The concentrations calculated below will be added in the revised paper.

For NO: (NO cylinder concentration) x (flow rate) x (time) / (chamber volume) = (54 ppm)(1 L/min)(20 min) / (9010 L) = 0.12 ppm in chamber

For NO2: (NO2 cylinder concentration) x (flow rate) x (time) / (chamber volume) = (80 ppm)(1 L/min)(20 min) / (9010 L) = 0.18 ppm in chamber

Elapsed time of t = 0 does not correspond to the first injection. Vertical lines have been added to Figure 3 to indicate the injection time. A copy of the revised Figure 3 is below and will be in the revised manuscript.

Figure 3: Injections and response (mixing) time for NO and NO2, done on different days, Vertical lines are injection times for NO and NO2. Injection time was five minutes each.

**REVIEWER COMMENT #8:** Line 262: The authors state that particles only become wellmixed in their chamber after 60 to 90 min. A figure showing how the aerosol number/volume concentration changes as a function of elapsed time after aerosol injection into the chamber should be used to back up their claim.

AUTHOR RESPONSE: A heat map, showing size as a function of time has been added to supplementary information and the figure (Figure 2) is included here, below. This is the same data shown in Figure 5 in the manuscript, but including earlier size distribution measurements. The coloring of the heat map, compared to Figure 5, is distorted because the highest concentration during mixing is ~7 times larger than the maximum in Figure 5. The figure below will be included as supplementary information in the revision.

---

## Author Comment (AC2) · 13 May 2018

**RESPONSES TO REVIEWER #1**

The authors thank the reviewer for his detailed and comprehensive comments that will help make the paper stronger. We address all the comments and suggested corrections as best as we can, below.

General:
**REVIEWER COMMENT #1**: *The authors constantly claim the novelty of their newly constructed chamber, yet fail to provide any experimental evidence that could help to identify any unique aspect of the current chamber setup. What new discoveries could potentially be produced by the NCAT chamber compared with all the other chambers that are also equipped to study BB* aerosols?

**AUTHOR RESPONSE**: The reviewer is right in indicating that there are numerous indoor smog chambers already constructed; however, we do not claim that the chamber itself is novel. What we consider unique and novel is the "Combustion-Chamber System". Given the unique combination of our custom designed and built instruments and combustion aerosol generation system, our entire chamber system is better described as "Unique". The following are unique qualities of our system: BB combustion and aerosol generation using a tube furnace to produce aerosols under controlled burning conditions (temperature, air flow, oxygen content, and amount of fuel burned) and the ability to clearly visually differentiate brown carbon (often formed at around 450-500 $^{o}$C), Black Carbon (formed around 650 - 750 $^{o}$C) and other forms (mixed brown/black carbon) at intermediate temperatures. Figure 1 shows the two extreme forms of carbon collected on filters. The generation and introduction of the soot particles into the chamber is described in line 194-197. The generation of soot particles and the introduction method, along with filter and impinger sampling and the integration of the chamber to the cavity ring-down/nephelometer system are not common arrangements, and we haven't come across chambers used in this manner. Many chambers are difficult to clean and therefore the characterization for wall loss and light intensity is a moving-target. Our chamber can more easily have each of the FEP panels cleaned and replaced. We believe the entire system is novel, not just the chamber. There is more to say here about the "novel" nature of the chamber system: We have characterized this chamber with particles that are combustion particles from biomass burning and have expended efforts to fully document the characterization with this manuscript. This adds to the "novel" or more rare nature of the chamber in that many chamber characterization efforts are undocumented, which makes the chamber results suspect and uncertain. Other chambers have been characterized with simple experiments, which are incomplete or unsatisfactory for conducting experiments later with different particles, such as from biomass burning. These aspects of this chamber and Figure 1 will be included in the revised manuscript.

[Figure]

Figure 1. Filter samples of Biomass Burning aerosols produced in a tube furnace (a) at 450 $^{o}$C b) at 800 $^{o}$C. (Note the 800$^{o}$C sample is highly concentrated)

**REVIEWER COMMENT #2:** *One novel aspect of this chamber, as the authors state in the conclusion section, is 'the use of a tube furnace ...., that enables the simulation of different burning stages ...'. However, the authors did not provide any experimental observations in terms of the physical, chemical, and optical properties of BB aerosols produced from different burning stages of a given fuel. Relevant measurements need to be added in the manuscript to support this claimed novelty of the chamber.*

**REVIEWER COMMENT #3:** *While the title highlights that the NCAT chamber is particularly suitable to study the aging processes of BB aerosols, the authors emphasize their focus on primary organic aerosols throughout of the main text. One illustration experiment that shows how photochemical aging would affect the optical properties of BB aerosols needs to be provided.*

**AUTHOR RESPONSE**: Comments #2 and #3 are related and the responses are combined. We will add preliminary measurements of average values of single scattering albedo (SSA) (Table 1) and SSA as a function of wavelength from 500 to 570 nm (Figure 4a, b, c) of BB aerosols produced by combusting African fuel samples (eucalyptus combusted in tube furnace at 500 $^{o}$C) in response to comments and to make this section relevant. The plots show the single scattering albedo calculated from measurement of extinction and scattering cross sections. The measurements were conducted by sampling the particles soon after they were introduced into the chamber, after aging for 48 hours in the chamber in the dark, and after aging for 10 hours in the chamber with the UV lamps on. The measurements were done for three size bins (mobility diameters of 200, 300 and 400 nm).

**Table 3. SSA Values for Eucalyptus BB aerosol combusted at 500 °C.**

| Particle Diameter (nm) | 200 | 300 | 400 | Peak Mobility Diameter (nm) |
|---|---|---|---|---|
| Fresh | 0.646 ± 0.009 | 0.660 ± 0.010 | 0.669 ± 0.011 | 131 |
| Aged (Dark) | 0.729 ± 0.028 | 0.712 ± 0.021 | 0.720 ± 0.029 | 322 |
| Aged (UV) | 0.877 ± 0.017 | 0.923 ± 0.016 | 0.960 ± 0.020 | 385 |

a)

[Figure]

b)

[Figure]

c)

Figure 2. Single Scattering Albedo of Biomass Burning Aerosol obtained by combusting eucalyptus in a tube furnace at 500 $^{\circ}$C. Measurement was done soon after introduction into the chamber and aging in the dark and with UV lamps on a) 200 nm particles b) 300 nm particles and c) 400 nm particles.

**REVIEWER COMMENT #4:** *The authors spend an entire paragraph in the introduction section discussing the need to investigate the effect of relative humidity on the evolution of BB*

*aerosols. Can relative humidity be well controlled in the NCAT chamber? If so, the humidity effect on the SMPS measured size distribution of BB aerosols needs to be given.*

**AUTHOR RESPONSE**: We wanted to emphasize the need to investigate the effects of relative humidity on optical and chemical properties of BB aerosols, since determining the extent to which RH impacts absorption and scattering is part of our future work. Aerosols exposed to high humidity will change their chemical, physical, and optical properties due to their increased water content. This process has an important impact on the particles' ability to scatter or absorb visible light, and is highly relevant to combustion in specific regions and seasons in Africa. Our current goal is establishing baseline measurements under dry 0% RH conditions only. Future work will investigate the change in optical properties as a function RH in the chamber.

**REVIEWER COMMENT #5:** *A large body of discussions in the main text, such as air purification, chamber flushing and cleaning, light spectra shown in Figure 6, and temperature profiles shown in Figure 8, have been well established routine chamber operation procedures for many years and can be moved to Supplement. Experimental evidence needs to be given in Section 3.5 to support the statement 'the setup in our laboratory will allow us to sample particles directly from and chamber, and measure their size distributions and optical properties as a function of aerosol age'. As suggested earlier, one experiment that illustrates changes in the optical properties of BB aerosols as a function of photochemical aging needs to be given.*

**AUTHOR RESPONSE**: The reviewer is correct in stating that there is nothing new in the chamber operation procedure that is included in this manuscript. However, each chamber is unique and uses different light sources that need to be described. Extra care is taken in the air purification system by using a zero-air generator to remove all possible atmospheric pollutants to keep a clean chamber at nearly 0% RH for baseline measurements. The importance of documenting officially in a publication the description of these "basic" parts or systems of the chamber is critical in establishing the confidence in the results of experiments. Furthermore, the characterization of these parts and describing and documenting the characterization procedures and routine is critical and is NOT always well established and should not be moved to a supplement.

We also agree with the reviewer that supporting data to show changes in optical properties and size distribution as a function of aging is needed. While this is the focus of the work in our lab and detailed analysis will be presented in future manuscripts, we have included data to show the functionality of the system and to support the claims we have made in this work. See response to comment #3 and #2.

**Specific:**

**REVIEWER COMMENT #6**: *Page 3, Line 87: Most static Teflon chambers are actually not ideal for studying aerosol aging. As the experiment proceeds, the Teflon bag volume will be continuously depleted, eventually leading to significant particle wall losses. A typical chamber experiment usually lasts for 24-36 hours, which equals to approximately one to two days of atmospheric OH exposure given the average OH concentration of ~106 molecules cm-3 generated in the chamber.*

**AUTHOR RESPONSE**: The chamber is operated as a "batch reactor" having a fixed volume of 9 m$^3$. The Teflon walls are somewhat flexible, allowing for small changes in volume during injections without a change in pressure. The reviewer is correct in that dark experiments generally happen over a 24 - 48-hour period, such as the ones performed by Kalogridis et al. (2018). The use of UV lights can allow some reactions to proceed faster. Our chamber can more easily have each of the FEP panels cleaned and replaced. The figures below show the chamber before it was covered with Masonite and some of the frames.

[Figure]

Figure 3. The NCAT Chamber. This can be added in the manuscript if the editor and the reviewer consider it useful.

The schematics of the individual FEP panel joints is shown in Figure 3.

[Figure]

Figure 4. The corner of an FEP frame panel

**Reference**:
A.-C. Kalogridis, O.B. Popovicheva, G. Engling, E. Diapouli, K. Kawamura, E. Tachibana, K. Ono, V.S. Kozlov, K. Eleftheriadis; "Smoke aerosol chemistry and aging of Siberian biomass burning emissions in a large aerosol chamber; Atmospheric Environment Volume 185, July 2018, Pages 15-28

**REVIEWER COMMENT #7:** *Page 5, Line 165: Measurements need to be provided in the Supplement to support the statement 'gas purity is less than ...and both methane and non-methane hydrocarbons'.*

**AUTHOR RESPONSE**: These specifications came from the manual for the zero-air generator. It is not independently verified. The text in the manuscript will be edited to reflect this.

**REVIEWER COMMENT #8:** *Page 7, Line 228-230: Have the authors tried to vary the atomizing pressure or use other solvents (e.g., acetonitrile) to dissolve impinger collected BB aerosols to see if these procedures would make a difference on the measured aerosol size distribution?*

**AUTHOR RESPONSE**
We thank the reviewer for this suggestion; however, this is outside the scope of this paper, as it has been shown elsewhere (Lewis et. al, 2009) that the structure of BB aerosol particles collapses under humid conditions. This is especially the case for impingement, which represents a supersaturated scenario.

Reference:
Lewis, K. A., Arnott, W. P., Moosmüller, H., Chakrabarty, R. K., Carrico, C. M., Kreidenweis, S. M., Day, D. E., Malm, W. C., Laskin, A., Jimenez, J. L., Ulbrich, I. M., Huffman, J. A., Onasch, T. B., Trimborn, A., Liu, L., and Mishchenko, M. I.: Reduction in biomass burning aerosol light absorption upon humidification: roles of inorganically-induced hygroscopicity, particle collapse, and photoacoustic heat and mass transfer, Atmos. Chem. Phys., 9, 8949-8966, 10.5194/acp-9-8949-2009, 2009.

**REVIEWER COMMENT #9:** *Page 8, Line 275-280: How was the onset of gas chamber losses defined here? It seems like over the entire course of gas chamber loss measurements, standard gases are continuously injected into the chamber and withdrawn out of the chamber with a balanced flow rate. If so, uncertainties caused by chamber mixing timescale need to be considered. To fully isolate the effect of dilution on the measured gas wall loss rate, the authors are suggested to inject standard gases, turn off the injection flow and wait till well mixed, and then start the wall loss measurement.*

**AUTHOR RESPONSE**: The reviewer is correct. Wall loss measurements did not begin until concentrations of the gas had peaked and leveled off. Figure 3 only shows the injections of the gas to show how much time elapses between injection of gas and stabilization in the chamber. Figure 3 will be updated to show these injection times as vertical lines on the graph as shown below.

[Figure]

Figure 4: Injections and response (mixing) time for NO and $NO_2$, done on different days, Vertical lines are injection times for NO and $NO_2$. Injection time was 5 minutes each

Figure 4 in the manuscript shows what a typical wall loss looks like after the chamber is well mixed.

**REVIEWER COMMENT #9:** *Page 10, Line 338: Initial particle number concentration might be an indicator of the particle pysicochemical properties, e.g., phase state. Understanding the relationship between initial particle number concentration and the loss rate needs to be included in the current study.*

**AUTHOR RESPONSE**: Certainly, it would be great to characterize for a range of sizes, and that can be suggested as future work. For this first manuscript, our reported work should be an adequate start, given the stated goals of the current project. There are more results to be published based on measurements being conducted on fuel sources from Africa. The size we picked is relevant, given the size distribution of combustion particles and availability of data

from cited references, which only provide wall loss for 100 nm particles. Also see response to comment # 11)

**REVIEWER COMMENT #10:** *Page 10, Line 346: When BB aerosols produced from a 500 °C furnace are introduced into a chamber operated at room temperature, organic vapors generated together with BB aerosols will undergo condensation on existing particles. On the other hand, organic compounds may evaporate from BB aerosols upon dilution in the chamber. So strictly speaking, the focus here is not POA.*

**AUTHOR RESPONSE**: The reviewer is correct in some sense. POA, previously treated as nonvolatile and nonreactive, can evaporate, oxidize, and re-condense to form SOA, which is known as aging of POA (Robinson, 2007). The cooling of particles upon introduction into the chamber can cause the semi volatile organics to condense on the particles. Due to the presence of semi- and low-volatile organics in the emissions, the freshly emitted particles from wood combustion are coated with organic matter (Tissari et al., 2008; Torvela et al., 2014). The coating often substantially increases due to the condensation of secondary organic matter formed either from anthropogenic or from biogenic organic precursor gases in the atmosphere (e.g., Akagi et al., 2012, Tiitta et al., 2016). So, in that sense the measured properties do not exactly represent fresh POA even in the field measurements. But, for measurements of fresh soot, the chamber is clean and free from typical atmospheric pollutants. The initial measurements are the closest we can get to POA before aging takes place, while not sampling hot, concentrated gases. Measuring the wall loss rate for organic compounds would be difficult with the current suite of available instrumentation, so it has not been performed in this work. However, we are collaborating with Dr. Surratt's Lab at UNC Chapel Hill for chemical characterization of filter samples collected at different stages. The results will be reported in upcoming publications.

**References:**
Robinson, A. L. *et al.* Rethinking organic aerosols: Semivolatile emissions and photochemical aging. *Science* **315**, 1259–1262, doi: 10.1126/science.1133061 (2007).

Tissari, J., Lyyränen, J., Hytönen, K., Sippula, O., Tapper, U., Frey, A., Saarnio, K., Pennanen, A. S., Hillamo, R., Salonen, R. O., Hirvonen, M.-R., and Jokiniemi, J.: Fine particle and gaseous emissions from normal and smouldering wood combustion in a conventional masonry heater, Atmos. Environ. 42, 7862–7873, 2008.

Torvela, T., Tissari, J., Sippula, O., Kaivosoja, T., Leskinen, J., Virén, A., Lähde, A., and Jokiniemi, J.: Effect of wood combustion conditions on the morphology of freshly emitted fine particles, Atmos. Environ. 87, 65–76, doi:10.1016/j.atmosenv.2014.01.028, 2014

Akagi, S. K., Craven, J. S., Taylor, J. W., McMeeking, G. R., Yokelson, R. J., Burling, I. R., Urbanski, S. P., Wold, C. E., Seinfeld, J. H., Coe, H., Alvarado, M. J., and Weise, D. R.: Evolution of trace gases and particles emitted by a chaparral fire in California, Atmos. Chem. Phys., 12, 1397–1421, doi:10.5194/acp-12-1397- 2012, 2012

Tiitta, P., Leskinen, A., Hao, L., Yli-Pirilä, P., Kortelainen, M., Grigonyte, J., Tissari, J., Lamberg, H., Hartikainen, A., Kuuspalo, K., Kortelainen, A.-M., Virtanen, A., Lehtinen, K. E. J.,

Komppula, M., Pieber, S., Prévôt, A. S. H., Onasch, T. B., Worsnop, D. R., Czech, H., Zimmermann, R., Jokiniemi, J., and Sippula, O.: Transformation of logwood combustion emissions in a smog chamber: formation of secondary organic aerosol and changes in the primary organic aerosol upon daytime and nighttime aging, Atmos. Chem. Phys., 16, 13251-13269, https://doi.org/10.5194/acp-16-13251-2016, 2016.

**REVIEWER COMMENT #11:** *Page 23, Figure 5: Please also provide the measured particle wall loss rate as a function of the particle diameter.*

**AUTHOR RESPONSE**: We have demonstrated that 100 nm particles (those with the shortest chamber lifetime, and far smaller than we plan to use for optical characterization) persist for sufficient periods to enable their collection and/or optical measurement. We have not presented size-dependent wall loss rates constants in this work, nor have many others. The stated goal of this work is optical property characterization. If we were to determine SOA yield or combustion emission factors, such size-dependent wall loss rate constants would be necessary, and we would perform that level of characterization should we expand to those goals. However, it is not necessary for our current purposes. In short, there is no step in our data analysis where a particle wall loss rate constant would be used, so measurement of size-dependent wall loss rate constants is not currently useful. However, the comments may be significant if only the reviewer explained why he thinks the amount of characterization results are insufficient. The reviewer has not specifically stated what they feel is missing from the manuscript.

**REVIEWER COMMENT #12:** *Page 27, Table 1: What type of detector is attached to GC for the identification of hydrocarbons?*

**AUTHOR RESPONSE**: The detector is a Flame Ionization Detector, and this will be inserted in the revision. Other details will be included. A dual column isn't a typical configuration, and should be included.